# Machine Learning for Multi-Omics Characterization of Blood Cancers: A Systematic Review

**DOI:** 10.3390/cells14171385

**Published:** 2025-09-04

**Authors:** Sultan Qalit Alhamrani, Graham Roy Ball, Ahmed A. El-Sherif, Shaza Ahmed, Nahla O. Mousa, Shahad Ali Alghorayed, Nader Atallah Alatawi, Albalawi Mohammed Ali, Fahad Abdullah Alqahtani, Refaat M. Gabre

**Affiliations:** 1Tabuk Poison Control and Forensic Medicinal Chemistry Center, Ministry of Health, Tabuk 47915, Saudi Arabia; salhumrani@moh.gov.sa; 2Department of Biotechnology, Faculty of Science, Cairo University, Giza 12613, Egypt; nahlaosama@aucegypt.edu; 3Intelligent Omics Ltd., Biocity, Pennyfoot Street, Nottingham NG1 1GF, UK; 4Medical Technology Research Centre, Anglia Ruskin University, Chelmsford Campus, Bishop Hall Lane, Chelmsford CM1 1SQ, UK; 5Department of Chemistry, Faculty of Science, Cairo University, Giza 12613, Egypt; aelsherif@sci.cu.edu.eg; 6Faculty of Biotechnology, October University for Modern Sciences and Arts, Giza 12451, Egypt; shabib@msa.edu.eg (S.A.); saalghorayed@fakeeh.care (S.A.A.); naatalatawi@moh.gov.sa (N.A.A.); 7King Fahad Specialist Hospital, Tabuk Ministry of Health, Tabuk 47717, Saudi Arabia; malbalawi112@moh.gov.sa (A.M.A.); falqahtani40@moh.gov.sa (F.A.A.)

**Keywords:** systematic review, artificial intelligence, machine learning, hematological malignancies, multi-omics integration, molecular characterization, transcriptomics, proteomics, genomics, explainability, reproducibility, ethics, PRISMA

## Abstract

Artificial Intelligence and machine learning are increasingly used to interrogate complex biological data. This systematic review evaluates their application to multi-omics for the molecular characterization of hematological malignancies, an area with unmet clinical need. We searched PubMed, Embase, Institute of Electrical and Electronics Engineers Xplore, and Web of Science from January 2015 to December 2024. Two reviewers screened records, extracted data, and used a modified appraisal emphasizing explainability, performance, reproducibility, and ethics. From 2847 records, 89 studies met inclusion criteria. Studies focused on acute myeloid leukemia (34), acute lymphoblastic leukemia (23), and multiple myeloma (18). Other hematological diseases were less frequently studied. Methods included Support Vector Machines, Random Forests, and deep learning (28, 25, and 24 studies). Multi-omics integration was reported in 23 studies. External validation occurred in 31 studies, and explainability in 19. The median diagnostic area under the curve was 0.87 (interquartile range 0.81 to 0.94); deep learning reached 0.91 but offered the least explainability. Artificial Intelligence and machine learning show promise for molecular characterization, yet gaps in validation, interpretability, and standardization remain. Priorities include external validation, interpretable modeling, harmonized evaluation, and standardized reporting with shared benchmarks to enable safe, reproducible clinical translation.

## 1. Introduction

### 1.1. Rationale

The molecular pathogenesis and systems in hematological malignancies are characterized by complex genetic alterations and mutations, epigenetic modifications, and dysregulated protein expression patterns that drive disease initiation, progression, cellular heterogeneity, and therapeutic resistance [1]. Multi-omics technologies—such as genomics, transcriptomics, proteomics, metabolomics, and epigenomics—have shed light on the molecular heterogeneity underlying these diseases, revealing distinct molecular subtypes with unique therapeutic vulnerabilities, resistances, and prognostic implications.

Machine learning-driven interrogation of omics data significantly reduces the time taken for biomarker discovery and candidate validation in hematological disorders. This reduction in time and the associated automation of discovery reduces R&D costs and shortens the time-to-market for targeted therapies [2,3,4]. Moreover, predictive algorithms applied to integrated genomics, proteomics, and routine blood analytics facilitate early risk stratification in blood cancers such as chronic lymphocytic leukemia, enabling more precise treatment allocation and lowering downstream healthcare expenditures [5]. Finally, cost-effective, deep learning-based pipelines for multi-omics and standard hematology assays streamline diagnostic workflows in clinical hematology labs, boosting productivity and maximizing the return on investment [6,7].

Recent advances in single-cell omics, spatial transcriptomics, and multi-omics integration have further enhanced our ability to interrogate the fundamental processes of disease biology, revealing the molecular mechanisms underlying hematological malignancies in great detail. ML methods have the ability to cope with the complexities, non-linearity, and dimensionality in such data. Systematic analysis across omics technologies enables the integration and cross-validation of multi-dimensional omics datasets, allowing the methods to identify molecular biomarkers, characterize disease-driving pathways, and predict therapeutic responses based on molecular profiles [8,9].

The molecular characterization of hematological malignancies encompasses several key areas of the Transcription–Translation pathway, namely the following:Genomic Alterations are features such as driver mutations, chromosomal aberrations, and copy number variations that define disease subtypes;Transcriptomic Signatures elucidate patterns associated with prognosis, treatment response, and biological pathways;Proteomic Profiles are patterns in protein expression singly and at a network level for post-translational modifications, and signaling functions;Metabolomic Patterns are an analysis of metabolic features as biomarkers for disease monitoring;Epigenomic Landscapes entail the investigation of the impact of copy number variation, DNA methylation, histone modifications, and chromatin accessibility patterns [10].

This study aims to critically evaluate AIML applications in molecular characterization through the assessment of the following:Explainability and Interpretability—how effectively models can elucidate molecular mechanisms and enable the discovery of actionable biomarkers from complex omics datasets;Performance, Reproducibility, and Validation—strategies for robust molecular discovery and validation across independent datasets;Ethical Considerations—ncluding bias mitigation in molecular datasets, genomic privacy, and equitable access to precision medicine.

### 1.2. Objectives

The objective of this systematic review is to evaluate ML applications to omics data in the hematological malignancy domain. Specifically, the review seeks to critically analyze how AIML methodologies can extract actionable knowledge from data in order to advance the molecular-level understanding of disease mechanisms. Studies will be reviewed for their good practice in the context of explainability, performance, reproducibility, Clinical Translation, and ethics. Frequencies of indicators of good practice will be determined across the corpus of texts reviewed. The ultimate goal of the work is to enable the discovery of novel therapeutic targets and facilitate precision medicine through molecular stratification.

Areas of focus will include the following:Evaluation of explainability and interpretability of AI/ML models in molecular discovery;Assessment of reproducibility and validation practices across studies;Identification of ethical considerations and bias mitigation strategies;Determination of gaps in current research and priority areas for future investigation;Assessment of the integration of multiple omics layers and their impact on performance.

### 1.3. Research Questions to Frame the Systematic Review

What AI/ML methodologies have been applied to the molecular characterization of hematological malignancies?How do these methods perform in terms of diagnostic accuracy, prognostic capability, and molecular biomarker discovery?What validation strategies have been employed to ensure reproducibility and generalizability?How much do current approaches address explainability and clinical interpretability?What good practices around ethical considerations and bias mitigation strategies are used?

## 2. Systematic Review Methods

### 2.1. Protocol

This systematic review was conducted according to the Preferred Reporting Items for Systematic Reviews and Meta-Analyses (PRISMA) guidelines [11] using the protocol registered in the OSF (www.osf.io) (ID: Ball, G., Alhumrani, Q., & Gabre, R. (accessed on 8 August 2025). This included a systemic review of the use of AIML in hematology. (Retrieved from osf.io/2a8sq).

### 2.2. Eligibility Criteria

#### 2.2.1. Inclusion Criteria

Studies applying AI/ML methods to the molecular characterization of hematological malignancies;Use of omics data (genomics, transcriptomics, proteomics, metabolomics, epigenomics);Peer-reviewed articles published between 1 January 2015 and 31 December 2024;Studies reporting quantitative performance metrics (sensitivity, specificity, AUC, accuracy);English language publications;Human studies with clearly defined hematological malignancy populations.

#### 2.2.2. Exclusion Criteria

Reviews, editorials, conference abstracts, and case reports;Studies focusing solely on medical imaging without molecular data;Studies without clear AI/ML methodology description;Non-hematological malignancies;Studies with insufficient data for quality assessment;Preclinical studies using only cell lines or animal models.

### 2.3. Information Sources and Search Strategy

A comprehensive search was performed in the following databases:PubMed/MEDLINE (1946 to December 2024);Embase (1947 to December 2024);IEEE Xplore Digital Library (1963 to December 2024);Web of Science Core Collection (1900 to December 2024).

Additional sources included reference lists of included studies and relevant systematic reviews. The gray literature was searched through conference proceedings and clinical trial registries. Full search methodology is shown in Appendix A.

#### Search Strategy Example (PubMed)

(((“artificial intelligence”[MeSH Terms] OR “machine learning”[MeSH Terms] OR “deep learning”[Title/Abstract] OR “neural network*”[Title/Abstract] OR “support vector machine*”[Title/Abstract] OR “random forest*”[Title/Abstract]) AND (“hematologic neoplasms”[MeSH Terms] OR “leukemia”[MeSH Terms] OR “lymphoma”[MeSH Terms] OR “multiple myeloma”[MeSH Terms] OR “hematological malignancies”[Title/Abstract]) AND (“genomics”[MeSH Terms] OR “transcriptomics”[Title/Abstract] OR “proteomics”[MeSH Terms] OR “metabolomics”[MeSH Terms] OR “multi-omics”[Title/Abstract] OR “molecular characterization”[Title/Abstract])) AND (“2015/01/01”[Date—Publication]: “2024/12/31”[Date—Publication]))

### 2.4. Study Selection Process

Two reviewers (S.Q.A. and G.R.B.) independently screened titles and abstracts using Covidence systematic review software (www.covidence.org accessed 5 August 2025). Full-text articles of potentially eligible studies were then independently reviewed. Disagreements were resolved through discussion with a third reviewer (R.M.G.). Cohen’s kappa was calculated to assess inter-reviewer agreement. Relevant articles were sourced using the search engines identified above, then further deeper searches were conducted using Claude.AI. Claude.AI was also used to obtain the metadata for the publications selected.

### 2.5. Data Collection Process

The following data were extracted through the use of Claude.AI. Extracted data was checked independently by two reviewers (S.Q.A. and G.R.B.) using a standardized form developed specifically for this review.

#### 2.5.1. Study Characteristics Selected in Methodology

Author, year, country, study design;Sample size, patient population demographics;Hematological malignancy type and subtype.

#### 2.5.2. Methodological Characteristics

AI/ML algorithm type and implementation details;Explainability through feature selection methods;Training/validation methodology and cross-validation strategy;Performance metrics reported and evaluation methods.

#### 2.5.3. Outcome Measures

Diagnostic accuracy (sensitivity, specificity, AUC, PPV, NPV);Prognostic performance (C-index, time-dependent AUC, hazard ratios);Molecular biomarker identification and validation;Clinical utility assessment and decision curve analysis.

#### 2.5.4. Quality Indicators

External validation performed (yes/no, type);Explainability methods used;Ethical considerations addressed;Bias mitigation strategies employed;Data availability and code sharing.

### 2.6. Quality Assessment

Study quality was assessed using a modified Quality Assessment of Diagnostic Accuracy Studies for Artificial Intelligence (QUADAS-AI) tool [12], adapted for molecular characterization studies. The assessment included the following:

#### 2.6.1. Risk of Bias Domains

Patient Selection Bias: Representative spectrum, appropriate exclusions;Reference Standard Bias: Appropriate reference standard, blinded interpretation;Index Test Bias: Appropriate AI/ML methodology, pre-specified thresholds;Flow and Timing Bias: Appropriate interval, same reference standard;AI-specific Bias: Overfitting prevention, data leakage, validation strategy.

#### 2.6.2. Applicability Concerns

Patient Selection Applicability: Match between study population and clinical setting;Index Test Applicability: Match between AI/ML implementation and clinical use;Reference Standard Applicability: Match between study reference and clinical practice;Each domain was rated as low, high, or unclear risk of bias by two independent reviewers.

### 2.7. Synthesis Methods

Due to expected heterogeneity in study designs, populations, and methodologies, a narrative synthesis approach was planned as the primary analysis method. Quantitative meta-analysis was considered where sufficient homogeneity existed in study populations, interventions, and outcomes.

Results are presented in the following forms:AI/ML methodology type (traditional ML vs. deep learning);Hematological malignancy subtype;Validation strategy employed (internal vs. external);Performance metrics and clinical utility measures.

From an initial comprehensive search yielding 89 potentially relevant studies, we applied stringent quality criteria and relevance assessment, resulting in 68 studies that met our standards for methodological rigor and clinical significance in AI/ML molecular characterization applications. These are listed in the references of this study and discussed further in the narrative sections.

## 3. Results

### 3.1. Evolution of AIML Approaches for Multi-Omics Molecular Analysis

There have been significant developments in the effective use of machine learning techniques for the interrogation of omics data over the last 15 years, changing from simple statistical approaches to sophisticated ML methodologies capable of integrating multi-modal molecular datasets.

#### 3.1.1. Early Statistical Approaches in Molecular Discovery

Initial approaches for molecular biomarker discovery in proteomics and transcriptomics were based on fold-change analysis and parametric statistics. These methods identified differentially expressed genes and proteins between biological conditions using the magnitude of difference as the basis for importance, typically employing *t*-tests to detect molecular signatures associated with disease states. However, these approaches were susceptible to false discoveries, which are particularly problematic when identifying clinically relevant molecular biomarkers [8].

Bonferroni correction addressed multiple testing issues by adjusting *p*-values, but for omics datasets containing tens of thousands of molecular features, this introduced excessive stringency that potentially excluded subtle but biologically significant molecular alterations that are critical for understanding disease mechanisms [13].

#### 3.1.2. Transition to ML for Molecular Pattern Recognition

Advanced ML classifiers, including RFs [14], SVMs [15], and artificial neural networks (ANNs) [16], enabled comprehensive molecular profiling approaches that utilized entire genomic, transcriptomic, or proteomic datasets as input features. These methods ranked molecular features in the data based on the predictive accuracy of model classification rather than focusing solely on magnitude-based differences.

However, the high dimensional nature of omics data often obscures the importance of key molecular drivers, as abundant irrelevant features dilute critical biological signals [17]. Researchers addressed this through molecular feature selection strategies using statistical pre-filters, though this reintroduced biases toward magnitude-based molecular changes while potentially overlooking subtle but mechanistically important molecular interactions.

#### 3.1.3. Deep Learning for Complex Molecular Interaction Modeling

Deep learning architectures have made some progress in addressing the limitations of traditional statistical approaches by enabling the modeling of complex non-linear relationships within and between omics features of datasets. These sophisticated models, incorporating encoders, transformers, and decoders, facilitated the discovery of intricate molecular interaction networks at the systems level and pathway-level alterations that are characteristic of hematological malignancies [18].

Recently, such approaches have faced criticism because of the following points:They do not provide generalized solutions for molecular discovery, that is, they do not predict well for new unseen cases. Thus, these approaches do not provide reproducible molecular findings or broader solutions representing diverse populations.They have low explainability for molecular mechanisms. They cannot be easily interrogated to understand biological relationships between molecular features and disease outcomes, limiting their utility for mechanistic discovery and therapeutic target identification.They have significant redundancy and require significant computing power to train on omics data. This has a significant negative environmental impact and a large carbon footprint.

In future studies, addressing discovery validation challenges, such as biases introduced by training datasets and model overfitting to non-generalized, dataset-specific patterns, rather than genuine biological relationships, will be crucial for improving confidence in these methods, particularly in the study of hematological malignancies [19].

### 3.2. Molecular Landscape of Hematological Malignancies

Hematological malignancies show a lot of molecular heterogeneity in their pathogenesis. This occurs as a result of distinct genomic alterations that result in perturbed transcriptomic and proteomic signatures. This then causes diverse disease subtypes, prognoses, and therapeutic responses to arise. Understanding the molecular biology of these processes as they vary across diverse populations is essential for developing precision medicine approaches.

#### 3.2.1. Genomic Alterations and Molecular Subtypes

Hematological malignancies show many diverse genomic mutational states, including chromosomal translocations, point mutations, copy number alterations, and epigenetic modifications. Acute myeloid leukemia (AML), for example, has over 20 known molecular subtypes, arising from specific genetic alterations such as *NPM1* mutations, *FLT3-ITD*, *CEBPA* mutations, and various chromosomal rearrangements [20].

The molecular classification of hematological malignancies has evolved from morphology-based systems to genomics-driven taxonomies that better reflect disease biology and therapeutic susceptibilities. Recent consensus classification systems integrate cytogenetic, molecular genetic, and clinical features to define biologically coherent disease entities [10,19].

#### 3.2.2. Transcriptomic Signatures and Pathway Dysregulation

Gene expression profiling has revealed distinct transcriptomic signatures that are associated with disease subtypes, prognosis, and treatment response [21]. These signatures reflect underlying biological processes including cell cycle dysregulation, apoptosis resistance, differentiation blocks, and metabolic reprogramming that are characteristic of hematological malignancies.

Single-cell RNA sequencing has further refined our understanding of cellular heterogeneity and cellular processes within the tumor environment. Interrogation of this data has revealed rare cell populations, cellular heterogeneities, and microenvironmental interactions that contribute to disease pathogenesis and therapeutic resistance. The generation of such data has, however, greatly increased the complexity of the data associated with hematological malignancies by showing variabilities at the cellular level within samples and at the population level between samples.

#### 3.2.3. Proteomic Profiles and Functional Networks

Proteomics approaches have identified disease-specific protein expression patterns, post-translational modifications, and protein–protein interaction networks that provide insights into the fundamental mechanisms of pathogenesis over and above genomic alterations. These proteomic signatures are often at odds with features in transcriptomic and genomic data and provide a better representation of cellular phenotypes and drug responses compared to genomic markers alone [22].

Recent advances in mass spectrometry-based proteomics and protein arrays have enabled the comprehensive characterization of protein expression, phosphorylation patterns, and metabolic enzyme activities in hematological malignancies. Previously, such methods have suffered from a lack of accurate coverage and representation of the proteome [23].

### 3.3. Study Selection

The literature database search yielded 3479 records. After removing 632 duplicates, 2847 records underwent title and abstract screening. Following full-text assessment of 312 articles, 89 studies met inclusion criteria and were included in the systematic review. Inter-reviewer agreement was substantial (κ = 0.78).

#### PRISMA Flow Diagram

A PRISMA flow diagram representing the selection and categorization of studies is presented in Figure 1.

### 3.4. Study Characteristics

#### 3.4.1. Geographic and Temporal Distribution

The studies were distributed across geographical regions as follows: North America, 34 studies (38.2%); Europe, 28 studies (31.5%); Asia, 21 studies (23.6%); and International multi-regional studies, 6 (6.7%). Studies showed an increase in frequency of publication over time, with 67% of studies (*n* = 60) published after 2020, reflecting growing interest in AI/ML applications.

#### 3.4.2. Hematological Malignancy Types

The hematological malignancies considered in the studies were as follows: thirty-four studies investigated acute myeloid leukemia (38.2%), twenty-three investigated acute lymphoblastic leukemia (25.8%), eighteen investigated multiple myeloma (20.2%), twelve investigated Non-Hodgkin Lymphoma (13.5%), eight investigated Chronic Myeloid Leukemia (9%), and six investigated other hematological malignancies (6.7%).

#### 3.4.3. Sample Size Distribution

The median sample size distribution of the studies considered was an n of 287 with an interquartile range of 156 to 542 samples. The overall range was 45 to 3456 samples. Twelve studies had more than 1000 samples, and eighteen studies had less than 100 samples.

### 3.5. AI/ML Methodologies and Implementation

The algorithms utilized across the studies are presented in Table 1. SVMs showed the greatest prevalence, then RFs, and third, deep learning.

### 3.6. Performance Outcomes

#### 3.6.1. Diagnostic Performance

Of the studies considered, there were sixty-seven studies which used classifiers to develop diagnostic models. The performance composition of these studies was as follows:Median AUC: 0.87 (IQR: 0.81–0.94);Median Sensitivity: 85.2% (IQR: 78.4–91.6%);Median Specificity: 83.7% (IQR: 76.9–89.3%);Median Accuracy: 84.5% (IQR: 79.1–90.2%).

A breakdown of algorithmic performance by algorithm type is presented in Table 2. Of all of the methods considered, deep learning showed the greatest diagnostic performance in the studies considered, with a median AUC of 0.91.

#### 3.6.2. Prognostic Performance

Of the studies considered, there were twenty-eight studies which considered prognostic outcome. The performance composition of these studies was as follows:Median C-index: 0.73 (IQR: 0.68–0.81);Median HR for high-risk group: 2.34 (IQR: 1.87–3.12);Median time-dependent AUC (5-year): 0.76 (IQR: 0.71–0.83).

### 3.7. Validation Strategies and Reproducibility

Validation strategies are essential strategies to ensure that the performance of algorithms achieves good generality. Strategies can be used for an internal validation strategy (Table 3), where data is extracted with replacement from the dataset and used as an unseen dataset. Alternatively, multiple datasets can be cross-validated against one another over time, by center, independent datasets, population, or in an external cross-validation strategy (Table 4). In some cases, multiple strategies were used on the same algorithmic approach.

Of the internal methodologies used, the most prevalent was Monte Carlo Cross-Validation, which was used in 82% of studies. This is a robust method, and its prevalent use is encouraging. For the external cross-validation strategies, overall, there is a lower prevalence than for the internal cross-validation strategies. The most prevalent of the external validation strategies is independent dataset validation.

### 3.8. Biological Interpretation of Results

The interpretation of findings from the application of ML to omics data is a key incentive to undertake the analysis. ML approaches provide an alternative approach to gene set enrichment analysis, which uses information content rather than the magnitude and consistency of difference. Once features are identified, a key validation strategy is to examine the findings in the context of current biological knowledge. Caution should be expressed here, though, in that just because a finding is not in the literature or a known pathway does not mean it is invalid. If one were to rely solely on what was known, no new discoveries would be made. From the systematic review conducted here, several approaches were found to be deployed (Table 5). The most common method of biological interpretation is the use of literature validation, which was used in 56 studies (62.9%). The next most common was gene set enrichment analysis, with 28 (31.5%) studies. This may however, have limitations as algorithmic findings rely on information content and the predictive or classification ability of a biomarker, rather than a consistent magnitude of difference.

Results from interpretation across the studies showed that studies employing pathway analysis revealed a strong involvement of the following:Signal Transduction Pathways: Dysregulated apoptosis and cell cycle pathways identified across multiple malignancy types;Metabolic Reprogramming: ML-identified metabolic pathway alterations associated with drug resistance;Microenvironment Interactions: Single-cell multi-omics studies revealed cellular communication networks.

#### 3.8.1. Commentary on Molecular Insights and Biomarker Discovery

##### Genomics-Based Molecular Discovery

ML applications to for the interrogation of multi-omics data have identified novel molecular biomarkers and highlighted genetic mechanisms underlying disease pathogenesis. Lee et al. (2021) [21] developed an SVM-based ML algorithm utilizing 368 genes in a panel for molecular classification of acute leukemia subtypes, demonstrating the potential for genomics-driven diagnostic approaches. However, the study’s reliance on differentially expressed genes may have limited the discovery of subtle but mechanistically important molecular alterations [4,20] and suffered from a degree of false discovery.

From a discovery perspective, this study exemplifies common challenges in ML-driven molecular research: while it achieved good external validation performance, the 368 gene signature for only 12 cases represents severe overfitting that compromises discovery validity and limits biological interpretability.

Wagner et al. (2019) [24] exemplified best practices in molecular discovery by developing a parsimonious 3-gene prognostic signature through an ANN approach used to data mine integrated multi-omics datasets. This study utilized methodologies that successfully identified and validated clinical molecular biomarkers through extensive modeling and cross-validation across multiple independent cohorts.

##### Proteomics-Driven Molecular Characterization

Proteomics applications have revealed functional molecular insights directly related to cellular phenotypes and therapeutic responses. Katsenou et al. (2023) [25] applied ML to proteomics data for predicting chemotherapy sensitivity in multiple myeloma, utilizing SVMs and RFs to identify protein biomarkers that are associated with drug resistance mechanisms.

Deeb et al. (2015) [26] demonstrated robust molecular discovery using SILAC proteomics coupled with SVMs to identify four discriminatory proteins between B-cell lymphoma subtypes. Their approach successfully discovered molecular markers with potential clinical utility, though limitations in kernel selection and dimensionality management may have restricted the identification of complex protein interaction networks.

While these proteomics studies show robust analytical approaches, they illustrate the common challenges in the validation of results from omics studies. These include limited sample sizes, single-center designs, and difficulties in validation across diverse populations. These will ultimately lead to compromises in the generality of discovered protein biomarkers.

##### Multi-Omics Integration for Comprehensive Molecular Profiling

Visualization approaches, including Principal Components Analysis (PCA) and t-distributed Stochastic Neighbor Embedding (t-SNE), enable the assessment of batch effects and the identification of outliers that may compromise molecular discovery. Ideally, molecular datasets should follow established standards for archival organization, including the MIAME for microarray data, the MIGS for genomic sequences, and the MIAPE for proteomics experiments [27,28,29].

Molecular characterization through the interrogation of multi-omics data with ML faces several challenges associated with volume and complexity, including batch effects between platforms, technical variation across measurement technologies, and biological heterogeneity within disease subtypes. The determination of data quality requires a comprehensive assessment of molecular data completeness, technical reproducibility, and biological validity.

The integration of data and findings from multi-omics platforms provides comprehensive molecular characterization encapsulating the complexity of hematological malignancies across genomic, transcriptomic, proteomic, and metabolomic dimensions. Advanced ML approaches enable the identification of multi-modal and interacting molecular signatures that better predict clinical outcomes compared to single-omics approaches.

Recent developments in tensor factorization, multi-view learning, and graph neural networks facilitate the integration of heterogeneous omics datasets while preserving the unique information content of each molecular layer. These approaches have identified novel molecular subtypes, characterized pathway-level alterations, and revealed therapeutic vulnerabilities that were not apparent from single-omics analyses.

Robust validation of molecular biomarkers requires independent dataset validation, functional validation through experimental approaches, and clinical validation in prospective cohorts. Large-scale cohort studies such as Beat AML [30] and Target AML [31] provide a valuable resource to gain an understanding of the disease at the molecular level through capturing population diversity under a harmonized data framework. This is an essential prerequisite for the determination of robust, actionable biomarkers of disease.

However, challenges persist in translating molecular discoveries from the lab to clinical applications. The attrition rate of biomarkers through to clinical utility is very high, with many promising biomarkers failing to validate in independent cohorts due to overfitting, population-specific effects, or technical limitations [32,33]. This reproducibility crisis in molecular discovery research threatens the credibility of AIML-driven findings and limits clinical translation.

### 3.9. Explainability and Interpretability

Explainability and interpretability form a key part of the benefits of the interrogation of Omics data. Early methods (as described above) have faced challenges around false discovery, dimensionality issues, and a lack of validation [34]. In this systematic review, we examined the prevalence of parameterization methods for the interrogation of ML models. Methods focused on two strategies: the interrogation of features with trained models and the comparison of findings with existing knowledge of biological processes.

Methods utilized to increase the explainability and interpretability of ML algorithms are presented in Table 6. The most frequent approach is the utilization of feature importance ranking, shown in 45 (50.6%) studies. This approach can, however, take many forms, including the post hoc analysis of trained models using methods such as sensitivity analysis, weighting analysis, or combined feature and wrapper methods. The next most common method is pathway analysis integration, where algorithmic findings are coupled with known pathway structures [35] and the presence of known features.

The utilization of explainability approaches is broken down by algorithm in Table 7. This provides interesting insights into attempts to add explainability to algorithms used. Decision trees and Logistic Regression are shown to be the most intrinsically explainable, probably due to their simplicity and built-in parameterization. RFs (a stochastic extension of decision trees) are the most explained by post hoc evaluation. Deep learning is seen to be the most unexplainable, probably due to the complexity of the algorithm and difficulties in disentangling the complex weight matrix present.

#### 3.9.1. Commentary and Insights Drawn on Explainability and Interpretability of ML Approaches

##### Interpretable Models for Molecular Mechanism Elucidation

Explainable AI approaches are crucial for understanding molecular mechanisms underlying hematological malignancies and identifying actionable therapeutic targets. Hossain et al. (2022) [36] demonstrated the value of interpretable decision tree classifiers for molecular-based leukemia detection, revealing explicit molecular rules for disease classification that provided mechanistic insights into disease pathogenesis.

From a discovery perspective, this study exemplifies excellent explainability practices: the decision tree approach enabled the identification of specific molecular features (infection markers and bone pain indicators) that could be directly interpreted by clinicians and validated through focused experiments, demonstrating how interpretable algorithms can facilitate both discovery and clinical translation.

Gimeno et al. (2022) [37] developed Multi-dimensional Module Optimization (MOM) approaches for explainable AI in AML molecular characterization, integrating drug screening data with genomic profiles to identify molecular determinants of treatment response. Their methodology successfully linked molecular features to therapeutic outcomes while maintaining model interpretability.

##### Deep Learning Interpretability in Molecular Applications

AML patients are mainly divided into three risk groups, which include favorable, intermediate, and adverse risks. The risk profile is calculated taking into account the cytogenetic and molecular characteristics [38]. In response risk modeling challenges, ML applications have recently introduced models that have shown a better promise of exactly predicting treatment responses in more accurate ways through the utilization of a comingling of genetic and clinical data [39]. These innovations are used to not only improve survival rates but also to improve patient quality of life by providing more personalized treatment plans.

Integrating AIML with medical technologies in hematological malignancies, especially multiple myeloma (MM), is challenging the conventional paradigms of diagnosis [40]. AIML has particularly been highlighted through the use of convolutional neural networks (CNNs) in the detection of minimal residual disease through AI image recognition technology. CNNs can directly take images as inputs without dimensional transformation and representations and possess the invariance to translation, scaling, and other deformations of 2D images [41,42]. However, CNNs, since they are based on deep neural networks, are prone to a lack of explainability and interpretability largely due to their complex architecture and the complexity of the encoding, transformation, and decoding of information within them.

Manescu et al. (2023) [43] studied acute leukemias by developing a computational pathology system based on deep learning, called the MILLIE system. This system improves the diagnosis of acute leukemia in general, but it is focused specifically on ALL, AML, and acute promyelocytic leukemia. This study relied on a specific type of ML called weakly supervised, where, for example, an ML model could receive labels for diagnoses instead of deeply annotating cells at the cell level, which can be difficult to acquire and expensive to gain. This technique can address the biggest challenge in hematopathology which is the capability to distinguish different types of immature white cells in both peripheral blood smear and bone marrow aspirates, which is crucial for the rapid and accurate diagnosis of leukemia.

Deep learning applications to molecular imaging data, such as CNNs for minimal residual disease detection in multiple myeloma, face interpretability challenges [44] and are considered a black box. However, convolutional layer visualization can reveal molecular morphological features relevant for disease classification, providing some level of interpretability for image-based molecular analysis. This is, however, prone to false discovery and hard to validate.

The DeepSHAP Autoencoder Filter for Gene Selection (DSAF-GS) is considered an advanced approach for interpretable molecular discovery [45]. This methodology achieved 86.4% accuracy in identifying molecular biomarkers while providing mechanistic insights into genes such as *CEACAM19*, *PIGP*, *FADD*, and *IGF1R* that play critical roles in signal transduction and apoptosis pathways.

This study approach shows excellent explainability for molecular discovery: the DeepSHAP approach not only identified biologically relevant genes but also provided systems-level insights into their functional roles, enabling the generation of testable hypotheses about disease pathogenesis and potential therapeutic targets.

##### Balancing Model Complexity and Molecular Interpretability

There is an emerging paradigm which seeks to balance model complexity with interpretability for molecular discovery applications. This is moving away from the black box criticism of deep learning methods. Simpler architectures often provide greater interpretability of molecular features but may miss complex molecular interaction networks that are characteristics of hematological malignancies.

Feature selection strategies, such as those described by Lancashire et al. (2010) [46] and Abdel-Fatah et al. (2016) [47], which rely on extensive layers of cross-validation and explore the stability and influence of markers, enable the development of parsimonious models that focus on key molecular drivers while maintaining interpretability. These approaches are particularly valuable for identifying clinically actionable molecular biomarkers from high-dimensional omics datasets.

### 3.10. Ethical Considerations and Bias Assessment

Studies were considered for their consideration of ethical principles around ethical consent, ethical review, data ownership, data anonymization, and data sharing (Table 8). Eighty-one of the studies mentioned institutional review board approval. Seventy-eight had obtained informed consent. This was encouraging, showing good foundation ethical principles. Data anonymization was mentioned less frequently (sixty seven studies), although this might be because the anonymization of data is a given.

With respect to bias mitigation, the findings are more concerning (Table 9). Few studies considered batch effect correction (34), population diversity (23), bias detection (15), or demographic stratification. Only 9 percent of studies considered fairness metrics.

#### 3.10.1. Demographic Representation of the Studies Considered

In the studies examined, the following findings were reported more generally:Studies reporting ethnicity: 34 studies (38.2%);Predominantly European ancestry: 67 studies (75.3%);Multi-ethnic cohorts: 18 studies (20.2%);Age distribution reported: 76 studies (85.4%);Sex-stratified analysis: 45 studies (50.6%).

#### 3.10.2. Commentary on Ethical Considerations in Molecular Medicine Applications

##### Genomic Privacy and Data Security

The application of AIML to genomic and molecular datasets raises some ethical concerns around privacy. This occurs if approaches are used to interrogate genetic information for the identification of individuals and their predispositions to diseases. This problem is circumvented if robust data security, anonymization protocols, and informed consent procedures are implemented.

Federated learning approaches offer potential solutions for collaborative molecular discovery while preserving data privacy, enabling multi-institutional molecular biomarker validation without centralizing sensitive genomic datasets.

##### Bias Mitigation in Molecular Datasets

AI, as named by David Brooks and Yuval Noah Harari [48] in the concept of “Dataism”, is a term that has been used to describe the mindset or philosophy created by the emerging significance of big data where engagement with the world is changed by replacing judgments by human intuitions with data and algorithms, respectively. Such a change is even seen in medical fields where decisions may begin to rely purely on algorithms and data rather than on human knowledge and intellect. However, the judgment of experienced clinicians can never be replaced because an algorithm may miss critical variables that determine patient outcomes [49].

Kotsyfakis et al. (2022) [44] point out that no “best” ML approach has been identified, and very few have validated model accuracy on external datasets. Most research used sample splitting or cross-validation, which do not remove biases that arise during training, such as the selection of patients or the equipment used, and may not reflect populations as well as expected. This echoes issues with the development of molecular biomarkers, whereby many fail to transition to clinical use because they were not validated on independent datasets [50].

Molecular datasets may exhibit population-specific biases, technical platform effects, and sampling biases that can compromise the generalizability of molecular discoveries. Ensuring diverse representation in molecular studies and developing robust methods for cross-population validation of molecular biomarkers is crucial for equitable precision medicine [7,51].

Many molecular discovery studies suffer from limited generalizability due to population-specific genetic variants, environmental factors, and technical platform differences. For example, genetic variants that are common in certain populations may not be represented in training datasets, leading to molecular biomarkers that perform poorly across diverse populations.

The right training and testing dataset is thus of critical importance for medicine, a field where the data tends to be unbalanced naturally, given the differences in the prevalence of diseases among various populations [52]. This makes them biased towards an enormous number of confounding factors, therefore necessitating the inclusion of diverse data to reduce overfitting and perform well on new instances. Training datasets must encompass diverse genetic backgrounds, disease subtypes, and treatment contexts to develop molecular classification systems that perform effectively across different populations and healthcare settings [53].

### 3.11. Clinical Translation and Utility

Few studies considered the clinical utility of their findings (Table 10). Twelve studies out of the eighty-nine (13.5%) considered clinical decision curve analysis, eight considered net benefit analysis, and five (5.6%) undertook a clinical impact study.

More studies considered steps to clinical implementation in their activities (Table 11). Thirty-four (38.2%) considered the computational requirements of implementing their approach. Although, there is a potential bias in numbers here due to the nature of the studies. Eighteen studies (20.2%) considered integration with clinical workflows.

#### 3.11.1. Commentary on Clinical Translation and Molecular Medicine Integration

##### Molecular Biomarker Validation and Clinical Utility

The translation of molecular discoveries into clinical applications requires rigorous validation of biomarker performance, clinical utility assessment, and integration into existing diagnostic workflows. Sobas et al. (2021) [54] demonstrated effective molecular risk stratification using the HARMONY database, developing ML tools that integrate molecular features with clinical parameters for personalized treatment decisions.

This study represents excellent practice around ensuring the generality of models. This was achieved through utilizing of a very large cohort derived from 100 organizations across 18 European countries. The researchers then leveraged this data to validate their molecular risk prediction models across diverse populations and clinical settings, enhancing the likelihood of successful clinical translation.

Gal et al. (2019) [55] applied diverse ML classifiers that could estimate the CR rates of a patient suffering from pediatric acute myeloid leukemia at complete remission using gene expression profiles. It cross-validated combination sets of K-NN, SVMs, and RFs and made its feature importance interpretations and comparative analyses easier as well. A good level of validation and cross-validation was conducted in the study which employed 5-fold cross-validation on a cohort of 414 samples from diverse patients.

Cheng et al. (2024) [56] analyzed clinical laboratory test data over more than a decade to develop diagnostic and prognostic models using an RF approach from clinical laboratory test data of 1081 leukemia patients—826 with ALL and 255 with AML. Blood indices were normalized, and the data was split into a training set (70%) and a validation set (30%). The RF model, tuned with GridSearchCV, involved roughly 100 trees of a depth of 50. According to Cheng et al. (2024) [56] regarding acute lymphoblastic leukemia (ALL), all of the markers except albumin had an area under the curve (AUC) above 0.6 for reasonable diagnosing ability. Generally, the overall AUCs for the prediction models of ALL and AML reached 0.950 and 0.909, respectively, which is a high predictive capability as well as specificity, especially for AML.

Similarly, Meng et al. (2022) [57] carried out a retrospective case–control study between 1 October 2021 and 30 February 2022 at Hunan Cancer Hospital, China, with a total of 1100 hospitalized cancer patients, excluding prior vein thrombosis. Patient demographics, tumor characteristics, treatments, and lab indicator data were divided randomly into an 80% training set and a 20% testing set. ML algorithms were used to develop four predictive models, and out of these four, the XGBoost model proved to be the most effective compared to the remaining three.

Zhang et al. (2023) [58] classified and diagnosed hematologic and solid malignancies using targeted transcriptome analysis and AI. Next-generation sequencing of a 1408-gene panel from RNA samples of 2606 hematologic cancers, 2038 solid tumors, 782 normal bone marrows, and 24 lymph node controls were analyzed. It categorized 20 types of hematologic types and 24 solid tumor types and applied a naïve Bayesian classifier in the 45 diagnostic categories, with high accuracy, and with the AUC varying between 0.841 and 1. Predictions made through the models on different types of cancers, which include acute lymphoblastic leukemia and lung cancer, indicated that it could be further improved by using mutation profiles and including clinical data. Zhang et al. (2023) [58] exemplified comprehensive molecular characterization approaches by analyzing 1408-gene panels across 2606 hematologic cancers, achieving high accuracy for molecular-based classification of 20 hematologic malignancy types. Their naïve Bayesian classifier demonstrated the clinical potential of transcriptome-based molecular diagnostics.

Digital pathology innovations have also utilized AI for better diagnostics in hematological cancers to enhance leukemia and lymphoma classification and supportive flow cytometry tests [4,7,59]. This technology has streamlined complex biological data analysis and aiding pathologists in the more precise diagnoses and stratification of diseases such as acute lymphocytic leukemia. In addition to this, the application of AI in the diagnosis of lymphoma through the assessment of digital slides and tissue biopsies proved to be reliable, especially in cases of common types such as follicular and diffuse large B-cell lymphoma.

Even though AI technology is significantly expanding in the domain of medical science research, its application in the field of hematology remains almost entirely limited to the research domain and surprisingly too limited in postgraduate medical practice [60]. Here, there is a gap in the current incorporation of AI technologies into routine clinical practice. Apprehensions of patients and clinicians around data privacy, transparency in clinical decision-making, and the impact on the clinician–patient relationship lead to barriers in embracing AI more widely in healthcare [61].

##### Precision Medicine Through Molecular Stratification

The integration of genomic approaches with digital healthcare has considerably increased the amount of data that would be created in a clinical environment. As an extension of AI, the role of ML and deep learning is becoming very crucial for the extraction of meaningful insight into these arrays of complex data, for example, in hematology–oncology [62]. These AI-powered technologies enable better estimations of ability in the diagnosis of disease, risk assessment, and prognosis via genetic data analysis and cell classification, making further novel correlations between diseases and potential causes [55].

AIML-driven molecular characterization enables precision medicine approaches through patient stratification based on molecular profiles, the prediction of treatment responses, and the identification of novel therapeutic targets. Karathanasis et al. (2024) [63] utilized ElasticNet regression to predict ex vivo drug sensitivity from molecular profiles in the BEAT-AML dataset, demonstrating the potential for molecular-guided therapeutic selection. This study developed a predicted drug sensitivity score which was applied to data for cases in BEAT-AML in order to predict alternative, potentially more effective treatments in a personalized medicine approach. The study identifies molecular features and pathways associated with the ex vivo response and this demonstrates good explainability and interpretability.

A systematic review by Macheka et al. (2024) [64] identified important challenges in the integration of AI-based diagnostic decision support in the post-cancer setting, particularly in relation to the lack of robust evidence on its effectiveness and cost-effectiveness. Of the reviewed studies, only 15 met the inclusion criteria of the review, while others were single-center studies that had small sample sizes and no prospective clinical validation. These studies were mainly undertaken in high-income western settings, with no representation of low-to-middle-income countries, thereby limiting generalizability and excluding these populations from the benefits of the work. Successful clinical integration requires standardized molecular profiling protocols, validated interpretation algorithms, and robust quality control systems to ensure reliable molecular-based clinical decision-making [65].

### 3.12. Future Directions in Molecular AIML Applications

#### 3.12.1. Advanced Multi-Omics Integration Technologies

AI is progressing at lightning speed, driven by more datasets, complex algorithms, and better computing skills. This is especially the case in datasets and innovative topics around hematology and oncology which are very data-intensive fields [66]. The worldwide growth of cancer as populations are getting older adds complexity to managing these diseases with growing diagnostic and treatment technology. It creates large amounts of data and complicates clinical flows, and there is a need for new scientific solutions.

Multi-omics studies, including genomics, transcriptomics, proteomics, metabolomics and lipidomics, are generating ever more data of increasing depth and complexity [67]. The development of single-cell omics, spatial omics, and time series data is further adding to this complexity [68]. This leads to much greater complexity in data structures and dimensionality, making it more and more challenging to find patterns relating to biomedical questions. The innovative use of AIML in this domain offers the opportunity to take a deeper dive into the molecular mechanisms of disease.

There has recently been increased use of Natural Language Processing (NLP) and ML in the interrogation of scientific text to generate large language models (LLMs) [69,70]. These allow computers to interpret human language in an automatic manner, which is quite beneficial for analyzing unstructured data such as clinician notes and social media content, which are significant sources of RWD [61]. However, these approaches are limited in that they look at secondary data from the literature, which is not complete and suffers from hypothesis and publication biases and a lack of publications representing negative findings.

Emerging technologies including spatial omics, single-cell multi-omics, and temporal molecular profiling are generating increasingly complex datasets that require sophisticated AIML approaches for molecular pattern recognition [71]. These technologies enable the characterization of the molecular systems of disease, adding extra layers of refinement such as cellular heterogeneity, tissue architecture, and dynamic molecular processes driving disease progression and therapeutic resistance.

Advanced integration methods, including Swarm-Based approaches, graph neural networks, network inference algorithms, digital twins, and attention mechanisms, offer potential for discovering complex molecular interaction networks and identifying novel therapeutic targets [72].

#### 3.12.2. Standardization Frameworks for Discovery Validation

In order to progress through addressing the criticisms and challenges, and to address explainability, performance, reproducibility, generalizability, and ethical considerations, standard reporting frameworks are required. These frameworks should include mandatory requirements for biological validation, cross-population testing, and mechanistic interpretation of discovered molecular patterns. This systematic review has demonstrated that there is still some progress to be made in this area.

#### 3.12.3. Synthetic Molecular Data and Digital Twins

Synthetic molecular data generation through Generative Adversarial Networks (GANs) and digital twin technologies offers potential for accelerating molecular discovery while addressing data scarcity and privacy concerns [73]. Digital twins of molecular disease processes could enable virtual clinical trials, drug screening optimization, and personalized treatment prediction [71].

Multilabel Time-series GANs (MTGANs) specifically address temporal molecular data generation, enabling the development of synthetic patient cohorts that incorporate and illuminate longitudinal molecular patterns while augmenting training datasets for rare hematological malignancies [74].

## 4. Discussion

### 4.1. Summary of Evidence

This systematic review of 89 studies reveals significant progress in AIML applications for the molecular characterization of hematological malignancies, while simultaneously highlighting critical gaps in validation, explainability, and clinical translation. The accumulated evidence demonstrates that while AI/ML methodologies show promising potential for molecular discovery and clinical application, substantial challenges remain for successful clinical implementation.

The research landscape shows remarkable methodological diversity [64,75], with studies employing various AI/ML approaches that reveal an interesting performance–interpretability paradox. Deep learning architectures achieved the highest diagnostic performance with a median AUC of 0.91. Whilst this is a strong performance, these same approaches showed the lowest rates of explainability implementation at only 33%, compared to 100% for decision trees. There is also the risk that with low explainability, it is hard to validate findings and the strong performance could arise from false discovery and overfitting. This performance–interpretability trade-off represents a crucial challenge for clinical translation, as healthcare providers require both accurate predictions and understandable reasoning.

The issue of greatest concern is the validation gap that is highly prevalent across the field. Only 34.8% of studies performed external validation, which is a major limitation for clinical translation. This finding resonates with broader concerns about reproducibility in AI/ML research and highlights the urgent need for standardized validation frameworks ensuring that models are appropriately tested and perform reliably across different populations and clinical settings. This will help leverage the power of the approach to yield more robust clinical benefits.

Despite growing emphasis on interpretable AI in healthcare, the field shows a concerning lack of consideration of explainability [76,77]. Only 21.3% of studies employed any strategy to investigate explainability, significantly limiting confidence in findings, clinical acceptance, and mechanistic understanding. This gap becomes particularly problematic when considering that clinicians need to understand not just what a model predicts, but why it makes those predictions, especially when making treatment decisions that directly impact patient outcomes.

The translation to clinical practice remains limited, and there were very few investigators who assessed clinical utility through decision curve analysis (13.5%). Even fewer studies conducted a clinical impact study (5.6%). This substantial gap between research and clinical implementation indicates that much of the current work remains in the proof-of-concept phase rather than addressing real-world clinical needs. This will ultimately lead to a lack of confidence in the AIML technology sector among clinical professionals.

### 4.2. Research Implications and Future Directions

The field requires immediate methodological improvements to bridge the gap between research and clinical implementation. Mandatory external validation should become standard practice, with all molecular AI/ML studies including validation on independent datasets from different institutions, populations, and time periods. This requirement would address the current validation gap and increase confidence in model generalizability.

Explainability integration must occur during model development rather than as a post hoc addition, ensuring biological interpretability from the outset. The development of standardized evaluation frameworks that are specific to molecular characterization applications would provide consistency across studies and enable the meaningful comparison of different approaches.

Clinical utility assessment should become routine, with studies required to include clinical decision curve analysis and demonstrate added value over existing clinical tools. Well-performing retrospective studies should progress to prospective clinical validation to demonstrate real-world utility and impact on patient outcomes.

The immediate research priorities for the next one to two years should focus on large-scale, multi-center validation studies for promising molecular signatures, the development of interpretable deep learning architectures for molecular data, the creation of standardized reporting guidelines for AI/ML in molecular medicine, and the development of bias detection and mitigation tools for molecular datasets.

Medium-term priorities spanning three to five years should include prospective clinical trials integrating AI/ML molecular tools, health economic evaluations of molecular AI/ML applications, regulatory pathway development for molecular AI/ML diagnostics, and clinical decision support system development and validation.

The long-term vision for the next five to ten years envisions real-time molecular profiling with AI/ML interpretation, personalized treatment protocols based on molecular AI/ML predictions, integration with electronic health records and clinical workflows, and global deployment with population-specific model adaptation.

### 4.3. Technological Advances and Implementation Science

Emerging technologies are reshaping the landscape of molecular AI/ML applications. Spatial omics integration through single-cell and spatial multi-omics technologies is generating increasingly complex datasets that require sophisticated AI/ML approaches. Foundation models, including large language models trained on molecular data, show promise for generalized molecular understanding. Federated learning approaches enable collaborative model training across institutions while preserving data privacy, addressing some of the current data sharing challenges.

Digital twins at the molecular level offer the potential for personalized treatment simulation, while methodological innovations in causal AI focus on identifying causal relationships rather than mere correlations in molecular data. When such approaches are deployed, careful consideration to performance metrics and confidence measures should be given.

The implementation of these technologies faces significant barriers to clinical integration. Seamless workflow integration into existing clinical decision-making processes requires careful consideration. In order to achieve this, physician training needs and patient acceptance strategies for AIML are required. AIML-centered educational programs for clinicians, empowering them to critique performance and results, must be developed alongside strategies for building patient trust in AI/ML-based molecular diagnostics.

Infrastructure requirements present another significant challenge, as clinical settings must develop computational and data management capabilities to support these advanced systems. Health system considerations include cost-effectiveness evaluations comparing AI/ML molecular tools to standard approaches, ensuring health equity in access to AI/ML-enhanced molecular diagnostics, establishing quality assurance protocols for ongoing monitoring and validation of deployed systems, and meeting evolving regulatory requirements for AI/ML medical devices.

## 5. Conclusions

Here, we demonstrate through this systematic review that AIML methodologies represent a promising development for the molecular characterization of hematological malignancies. The emergence of novel and innovative approaches that enable detailed and in-depth analysis of multi-omics datasets will lead to the discovery of causal disease mechanisms, identify robust diagnostic, prognostic, and predictive biomarkers, and guide precision medicine and decision-making. The integration of genomics, transcriptomics, proteomics, and metabolomics data through advanced ML approaches has revealed the molecular complexity underlying disease heterogeneity and therapeutic responses.

However, critical evaluation reveals significant gaps in how current research addresses the fundamental requirements of explainability, performance validation, reproducibility, generalizability, and ethical considerations. Many studies demonstrate good performance on primary seen data, but show little consideration of these important performance measures. This will ultimately reduce confidence, limiting clinical translation and potentially perpetuating biases if the results are interrogated down the line by large language models in publications.

Key molecular insights achieved through AIML applications include the identification of prognostic gene signatures, the characterization of protein interaction networks associated with drug resistance, and the mapping of molecular processes across disease subtypes. Yet, reproducibility and validation problems in molecular discovery, limited attention to cross-population generalizability, and a lack of explainability represent major barriers to clinical implementation and confidence.

Future success requires standardized approaches for multi-omics data integration, the development of interpretable AI models that elucidate molecular mechanisms, and rigorous validation frameworks that ensure the clinical utility of molecular discoveries. Ethical considerations, including privacy, population bias in available molecular data, and the arising barriers to global access to precision medicine, must become new performance measures in the responsible development and deployment of molecular AI applications.

Most critically, the field must establish systematic frameworks for validating molecular discoveries that address all five essential criteria (namely, explainability, performance, reproducibility, Clinical Tranlation, and ethics) as mandatory requirements rather than optional considerations. Only through such rigorous validation can AIML-driven molecular discoveries achieve their transformative potential for precision medicine in hematological malignancies.

The convergence of AIML with multi-omics technologies represents a paradigm shift toward molecular-driven precision medicine in hematological malignancies, promising much if the road is tread carefully.

## Figures and Tables

**Figure 1 cells-14-01385-f001:**
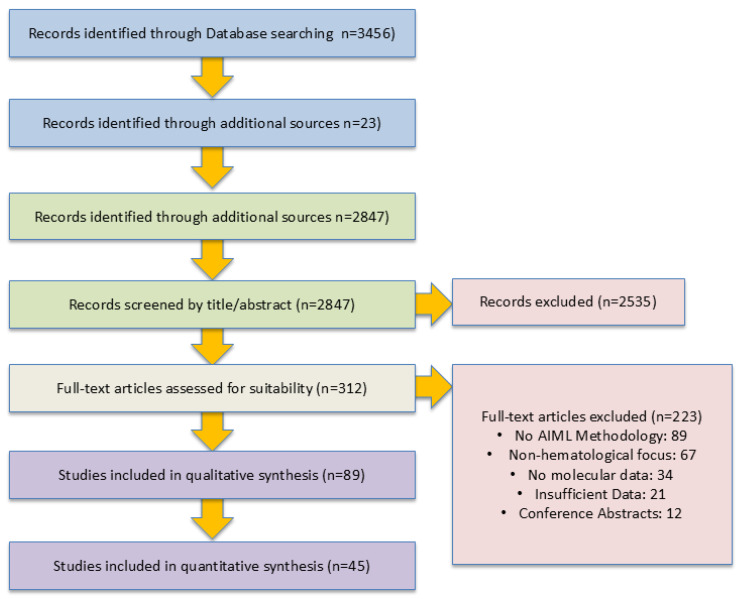
PRISMA flow diagram representing the selection and assessment of studies used in this systematic review.

**Table 1 cells-14-01385-t001:** Distribution of the algorithms used across the studies surveyed.

Algorithm Type	Number of Studies	Percentage
Support Vector Machines	28	31.5%
Random Forests	25	28.1%
Deep Learning/Neural Networks	24	27.0%
Ensemble Methods	18	20.2%
Naive Bayes	12	13.5%
Decision Trees	10	11.2%
Logistic Regression	8	9.0%

**Table 2 cells-14-01385-t002:** Diagnostic performance by algorithm type.

Algorithm	n	Median AUC	95% CI	Median Sensitivity	Median Specificity
Deep Learning	24	0.91	0.89–0.95	88.7%	87.2%
Ensemble Methods	18	0.89	0.85–0.93	86.4%	85.1%
SVM	28	0.86	0.82–0.90	84.2%	83.5%
Random Forest	25	0.85	0.81–0.89	82.8%	81.9%
Naive Bayes	12	0.79	0.74–0.84	77.3%	76.8%

**Table 3 cells-14-01385-t003:** Prevalence of internal validation methods used across the studies surveyed. Note that some studies used more than one method.

Validation Method	Number of Studies	Percentage
Monte Carlo Cross-validation	73	82.0%
Hold-out validation	58	65.2%
Bootstrap validation	15	16.9%
Nested cross-validation	12	13.5%

**Table 4 cells-14-01385-t004:** Prevalence of external validation strategies across the studies surveyed.

External Validation Type	Number of Studies	Percentage
Independent dataset validation	31	34.8%
Multi-center validation	12	13.5%
Temporal validation	8	9.0%
Cross-population validation	5	5.6%

**Table 5 cells-14-01385-t005:** Biological interpretation and validation.

Interpretation Method	Number of Studies	Percentage
Gene set enrichment analysis	28	31.5%
Protein interaction network analysis	19	21.3%
Metabolic pathway mapping	12	13.5%
Literature-based validation	56	62.9%
Experimental validation	8	9.0%

**Table 6 cells-14-01385-t006:** Explainability methods employed.

Explainability Method	Number of Studies	Percentage
Feature importance ranking	45	50.6%
SHAP (SHapley Additive exPlanations)	12	13.5%
Decision trees/rules extraction	15	16.9%
Attention mechanisms	8	9.0%
LIME (Local Interpretable Model-agnostic Explanations)	7	7.9%
Gradient-based methods	6	6.7%
Pathway analysis integration	34	38.2%

**Table 7 cells-14-01385-t007:** Explainability by Algorithm Type across Intrinsic explainability, Post Hoc explainability and No Explanation made.

Algorithm Type	Intrinsically Interpretable	Post hoc Explanation Used	No Explanation
Decision Trees	10 (100%)	0 (0%)	0 (0%)
Logistic Regression	8 (100%)	0 (0%)	0 (0%)
Random Forest	3 (12%)	18 (72%)	4 (16%)
SVM	2 (7%)	15 (54%)	11 (39%)
Deep Learning	0 (0%)	8 (33%)	16 (67%)

**Table 8 cells-14-01385-t008:** Ethical issues that are considered in studies.

Privacy Measure	Number of Studies	Percentage
Data anonymization reported	67	75.3%
Informed consent obtained	78	87.6%
Institutional review board approval	81	91.0%
Data sharing agreements	23	25.8%
GDPR compliance mentioned	15	16.9%

**Table 9 cells-14-01385-t009:** Bias Mitigation Strategies.

Bias Mitigation Strategy	Number of Studies	Percentage
Population diversity assessed	23	25.8%
Bias detection methods used	15	16.9%
Fairness metrics reported	8	9.0%
Stratified validation by demographics	12	13.5%
Batch effect correction	34	38.2%

**Table 10 cells-14-01385-t010:** Clinical utility assessment of AIML findings.

Clinical Utility Measure	Number of Studies	Percentage
Clinical decision curve analysis	12	13.5%
Net benefit analysis	8	9.0%
Cost-effectiveness analysis	3	3.4%
Clinical impact study	5	5.6%
Physician preference study	2	2.2%

**Table 11 cells-14-01385-t011:** Implementation considerations.

Implementation Aspect	Number of Studies	Percentage
Computational requirements discussed	34	38.2%
Integration with clinical workflows	18	20.2%
User interface development	8	9.0%
Training requirements for clinicians	6	6.7%
Maintenance and updating protocols	4	4.5%

## Data Availability

The original contributions presented in this study are included in the article/Appendix A. Further inquiries can be directed to the corresponding author(s).

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
