# Peer review of "Machine Learning for Multi-Omics Characterization of Blood Cancers: A Systematic Review"

_cells, 2025, doi:10.3390/cells14171385_

Round 1

Reviewer 1 Report

Comments and Suggestions for Authors

The study presents the results of several studies. It seems reasonable to transform this review into a scoping review, demonstrating the method of selecting published multi-omics studies that apply machine learning with validation models in hemato-oncology, in line with PRISMA.

The study authors have limited experience in the topic when analyzing the cited studies. 

There are no figures in the text.

The study content is engaging and highlights issues in the implementation of multi-omics, revealing distinct, highly variable patterns.

Minore:

Title: Please consider a change from generality to clinical transition

line 135 rather pathogenesis than etiology

Author Response

I would like to thank the reviewer for their positive comments.

In response

Comment 1:  The study presents the results of several studies. It seems reasonable to transform this review into a scoping review, demonstrating the method of selecting published multi-omics studies that apply machine learning with validation models in hemato-oncology, in line with PRISMA.

Response 1:  The study has been modified to a systematic review in line with PRISMA guidelines.  The review has been conducted in line with the broad areas of the title in hematological malignancies.  Review criteria have been set and a systematic review protocol has been defined..  A PRISMA flow chart has been added, and results are reported in the form of tables throughout the manuscript.  The original commentary has been maintained and merged with the review results.

Comment 2:  The study authors have limited experience in the topic when analyzing the cited studies. 

Response 2:  The authors are not experts in hematology but have extensive experience in the interrogation of multiomics data using machine learning.

Comment 3: There are no figures in the text. 

Response 3:  Tables representing the review results and a figure of the PRISMA flow diagram have now been added.

Comment 4:  The study content is engaging and highlights issues in the implementation of multi-omics, revealing distinct, highly variable patterns.

Response 4: Many thanks for your supportive comment.

Minore:

Comment 5:  Title: Please consider a change from generality to clinical transition

Response 5:  The title has been changed to reflect this request.

Comment 6:  line 135 rather pathogenesis than etiology

Response 6:  This has been changed in the text.  Due to additions around the PRISMA review this change is now on line 266.

Reviewer 2 Report

Comments and Suggestions for Authors

I find some weak points in this manuscript, which is interesting to read but, reading it with a reviewer's task, leaves some elements of doubt as to its overall quality. 
These weak points must be improved to make the manuscript acceptable from my point of view.

I find a strange reference to "the acoustic computational pathology system" (see line 314) that is not matching with the quoted article (ref. 53). This observation casts doubt on the quality of the reading and interpretation by the Authors of the cited bibliography, with the possibility that it was interpreted by an automated manner and sometimes with erroneous results.

Bibliographical references in a review must be relevant and such as to prove the source of the various statements presented. However, only one bibliographic reference appears on page 2 of the manuscript, and likewise there are paragraphs with statements without reference. Finally, there are references from preprints (as arXiv preprint), not relevant being not peer-reviewed publications.

Another note concerns abbreviations: these appear unevenly throughout the text, giving the impression that different parts were written separately and there was no proper standardisation of the final assembled draft of the manuscript. The authors' instructions indicate precise ways of how they are to be used, and these are fairly standardised.

Bibliographic citations also lack standardisation in the indication of the DOI code.

Author Response

I thank the reviewer for their constructive comments.  Please find below my responses.

Comment 1.  I find a strange reference to "the acoustic computational pathology system" (see line 314) that is not matching with the quoted article (ref. 53). This observation casts doubt on the quality of the reading and interpretation by the Authors of the cited bibliography, with the possibility that it was interpreted by an automated manner and sometimes with erroneous results.

Response 1:  I can assure the reviewer that all of the references have been checked and reviewed thoroughly in this manuscript.  I have chacked the DOI for the study and it tracks with the intended target, namely Manescu et al in the Cornell archive.  DOI: https://doi.org/10.1038/s41598-023-29160-4   

Comment 2:  Bibliographical references in a review must be relevant and such as to prove the source of the various statements presented. However, only one bibliographic reference appears on page 2 of the manuscript, and likewise there are paragraphs with statements without reference. Finally, there are references from preprints (as arXiv preprint), not relevant being not peer-reviewed publications. 

Response 2:  Four references are from ARXiv preprints.  25, 42, 53 and 63   Peer-reviewed full paper citation details have been retrieved for all references and these have been added to the manuscript.  The study has now been rewritten as a systematic review references added where context is needed.

Comment 3:  Another note concerns abbreviations: these appear unevenly throughout the text, giving the impression that different parts were written separately and there was no proper standardisation of the final assembled draft of the manuscript. The authors' instructions indicate precise ways of how they are to be used, and these are fairly standardised.

Response 3:  Abbreviations have been reviewed through the manuscript and standardised.  The reviewer was correct in their assertion that different parts of the manuscript were written by different individuals.  This is often the case in multidisciplinary collaborative research.  We have utilised the submission template in order to standardise the use of abbreviations throughout the manuscript

Comment 4:  Bibliographic citations also lack standardisation in the indication of the DOI code.

Response 4:  This has been corrected in the text.

Reviewer 3 Report

Comments and Suggestions for Authors

The authors undertook a review which examined the application of Artificial Intelligence (AI) and Machine Learning (ML) methodologies for molecular characterization of hematological malignancies through comprehensive multi-omics data integration. The review has no method section and has to be redesigned as a systematic review. The terms used are not explained and a glossary is required to explain for example: fold-change analysis, and parametric statistics , false discoveries, excessive stringency, predictive accuracy of model classification,  overlooking subtle but mechanistically important molecular interactions, non-linear relationships ,dimensionality reduction,  The least absolute shrinkage and selection operator, random forest graphsupport vector machine, and decision tree are reported to be the most potential algorithms
applied in this procedure, batch effects, Principal Components Analysis (PCA) or t-distribute stochastic neighbor embedding (tSNE) plot, interpretable decision fee,Deep Learning Interpretability in Molecular Applications, Molecular Interpretability.

In the conclusions it is not clear what are the essential criteria  referred to in " address all five essential criteria as mandatory requirements rather than optional considerations".     

The authors need to tabulate and explain in much more details  trials which have used AI and machine learning for direct patient care and explain the details in numbers and with case studies of individual patients.        

Author Response

Comment 1:  The review has no method section and has to be redesigned as a systematic review.

Response 1:  The review has been rewritten as a systematic review under PRISMA guidelines a method section has been added describing the approaches used in the PRISMA review.

Comment 2:  The terms used are not explained and a glossary is required to explain for example: 

Response 2  A glossary of terms has been added from line 950.

Comment 3:  In the conclusions it is not clear what are the essential criteria  referred to in " address all five essential criteria as mandatory requirements rather than optional considerations".     

Response 3:  The essential criteria listed in the title have now been restated in the appropriate place in the conclusions lines 940 to 941

Round 2

Reviewer 1 Report

Comments and Suggestions for Authors

The study design was revised in accordance with the PRISMA guidelines. Multiple changes make the current version difficult to read. 

The added subchapters discussing future direction and implementation are well balanced.

The authors should add some sentences related to the economic aspect of omics implementation.

Author Response

I would like to thank the reviewer for their constructive review of this manuscript.

Comment 1:  The authors should add some sentences related to the economic aspect of omics implementation.

Response 1:  The following text has been added to the rationale section of the introduction.  "Machine learning–driven interrogation of omics data significantly reduces the time taken for biomarker discovery and candidate validation in hematological disorders.  This reduction in time and the associated automation of discovery reduces R&D costs and shortening time-to-market for targeted therapies [2]. Moreover, predictive algorithms applied to integrated genomics, proteomics, and routine blood analytics facilitate early risk stratification in blood cancers such as chronic lymphocytic leukemia, enabling more precise treatment allocation and lowering downstream healthcare expenditures [3]. Finally, cost-effective deep learning–based pipelines for multi-omics and standard hematology assays streamline diagnostic workflows in clinical hematology labs, boosting productivity and maximizing return on investment [4]."

References

[2] Bourke, M., McInerney‐Leo, A., Steinberg, J., Boughtwood, T., Milch, V., Ross, A.L., Ambrosino, E., Dalziel, K., Franchini, F., Huang, L., Peters, R., Santos Gonzalez, F. & Goranitis, I. (2025). The Cost Effectiveness of Genomic Medicine in Cancer Control: A Systematic Literature Review, Applied Health Economics and Health Policy, 23(3), pp. 359–393. DOI: https://doi.org/10.1007/s40258-025-00949-w

[3] Tsagiopoulou, M. & Gut, I.G. (2024). Machine learning and multi-omics data in chronic lymphocytic leukemia: the future of precision medicine?, Frontiers in Genetics, 14, 1304661. DOI: https://doi.org/10.3389/fgene.2023.1304661

[4]  Obeagu, E.I., Ezeanya, C.U., Ogenyi, F.C., Ifu DD.  (2025). Big data analytics and machine learning in hematology: Transformative insights, applications and challenges., Medicine (Baltimore), 104(10), e41766. DOI: https://doi.org/ 10.1097/MD.0000000000041766

Reviewer 2 Report

Comments and Suggestions for Authors

The response about reference 53 and the phrase “acoustic computational pathology system” is insufficient; it does not clarify whether this was a typo or why there are references to “acoustic pathology” when referring to an article dealing with leukemia, or any other explanation to clarify the relationship of the term "acoustic computational pathology system" with the context of the manuscript. If the reviewer is puzzled about a sentence, so will the reader; authors must respond to the request of a reviewer by clarifying and, if anything, improving the sentence so that the reader can have a way of understanding.

Author Response

I would like to thank the reviewer for their comments.

Comment 1: it does not clarify whether this was a typo or why there are references to “acoustic pathology” when referring to an article dealing with leukemia, or any other explanation to clarify the relationship of the term "acoustic computational pathology system" with the context of the manuscript. 

Response 1:  I apologise for the confusion.  I thought the reviewer was querying the reference, not the text relating to the reference.  This was a typographical error.  The text has been modified to read.  "Manescu et al. (2022) studied acute leukemias by developing a computational pathology system based on deep learning"

Reviewer 3 Report

Comments and Suggestions for Authors

The authors have by transforming this review into a systematic review significantly improved their work and manuscript.

Author Response

I thank the reviewer for their helpful and constructive review.

Round 3

Reviewer 2 Report

Comments and Suggestions for Authors

No other comment.

Author Response

(The authors gave the same response as above.)
